# Leveraging Per-Instance Privacy for Machine Unlearning

**Nazanin Mohammadi Sepahvand** [* 1 2]  **Anvith Thudi** [* 3 4]  **Berivan Isik** [5]  **Ashmita Bhattacharyya** [3 4]
**Nicolas Papernot** [3 4]  **Eleni Triantafillou** [6]  **Daniel M. Roy** [4 7]  **Gintare Karolina Dziugaite** [8 2 9]

## Abstract

We present a principled, per-instance approach to quantifying the difficulty of unlearning via fine-tuning. We begin by sharpening an analysis of noisy gradient descent for unlearning (Chien et al., 2024), obtaining a better utility–unlearning trade-off by replacing worst-case privacy loss bounds with per-instance privacy losses (Thudi et al., 2024), each of which bounds the (Rényi) divergence to retraining without an individual data point. To demonstrate the practical applicability of our theory, we present empirical results showing that our theoretical predictions are born out both for Stochastic Gradient Langevin Dynamics (SGLD) as well as for standard fine-tuning without explicit noise. We further demonstrate that per-instance privacy losses correlate well with several existing data difficulty metrics, while also identifying harder groups of data points, and introduce novel evaluation methods based on loss barriers. All together, our findings provide a foundation for more efficient and adaptive unlearning strategies tailored to the unique properties of individual data points.

## 1. Introduction

Machine unlearning aims to efficiently remove the influence of specific subsets of training data, called *forget sets*. The need for unlearning arises in many scenarios, such as handling requests for data deletion (Mantelero, 2013; Cooper

et al., 2024), mitigating the impact of poisoning attacks (Liu et al., 2024b), or updating out-of-date information.

In modern large-scale AI training regimes, exact unlearning (i.e., anything equivalent to retraining the model from scratch, without the forget set) is prohibitively expensive. To meet this challenge, a range of approximate unlearning methods have been developed.

Approaches based on ideas from differential privacy (DP) offer meaningful worst-case guarantees (Guo et al., 2020; Sekhari et al., 2021). These will be our focus. Unfortunately, for non-convex models, like neural networks, DP-based unlearning guarantees have so far come with higher error than their counterparts not designed to support unlearning, limiting their practical applicability. One of the key problems is the mismatch between the worst-case, data-agnostic nature of DP and the data-dependent nature of unlearning. Our work attempts to bridge this gap.

Alongside work on theoretical guarantees, research into heuristics has flourished. The aesthetics of such research is to combine strong utility with empirical evaluations on metrics inspired by DP-based notions of approximate unlearning. It is not uncommon, however, for state of the art heuristics to be caught out by new attacks, revealing that they do not meet stringent theoretical criteria (Hayes et al., 2024; Pawelczyk et al., 2024). These challenges highlight the need for methods that can balance strong theoretical guarantees with strong practical performance.

Towards understanding failure modes of unlearning, recent empirical work has shown that the behavior of unlearning varies considerably across individual data points (Zhao et al., 2024; Baluta et al., 2024). While prior work has attempted to incorporate these insights into methodology and evaluation, we lack a theoretical basis to 1) explain how individual data influence unlearning and 2) exploit this effect optimally.

In this work, we introduce a principled approach to estimating the unlearning difficulty of individual data points. Our approach applies to learning with noisy gradient descent. Using recent results in differential privacy with Rényi divergences (Thudi et al., 2024), our proposed measures—*per-instance privacy losses*—bound the Rényi divergence between a model trained with and without an individual

---

[*]Equal contribution  [1]Department of Electrical and Computer Engineering, McGill University, Montreal, Canada [2]Mila, Montreal, Canada [3]Department of Computer Science, University of Toronto, Toronto, Canada [4]Vector Institute, Toronto, Canada [5]Google, Mountain View, US [6]Google DeepMind, London, UK [7]Department of Statistical Science, University of Toronto, Toronto, Canada [8]Department of Computer Science, McGill University, Montreal, Canada [9]Google DeepMind, Toronto, Canada. Correspondence to: Gintare Karolina Dziugaite <gkdz@google.com>.

*Proceedings of the 42nd International Conference on Machine Learning*, Vancouver, Canada. PMLR 267, 2025. Copyright 2025 by the author(s).

data point. Critically, per-instance privacy losses can be efficiently estimated during training, based on the norms of gradient associated to individual data points.

Armed with per-instance privacy losses, we revisit Chien et al.'s 2024 theoretical analysis of noisy gradient descent as an unlearning scheme (coined "Langevin unlearning" a.k.a "noisy fine-tuning"), based on training without the forget set. Our analysis provides a theoretical foundation for understanding how the number of iterations needed to approximately unlearn scales with per-instance privacy losses. Empirical validation demonstrates that our estimates of per-instance privacy losses serve as reliable predictors of unlearning difficulty under noisy fine-tuning.

While noisy fine-tuning is not widely used, standard (noise-less) fine-tuning is a surprisingly effective and simple approximate unlearning approach, and serves as a building block for SOTA methods, such as L1-Sparse fine-tuning (Hayes et al., 2024; Liu et al., 2024a). We demonstrate the applicability of our theoretical insights, showing they extend empirically to standard fine-tuning-based unlearning, even in the absence of explicit noise injection. Empirically, we find that per-instance privacy losses continue to predict unlearning difficulty for fine-tuning across multiple approximate unlearning metrics, datasets, and architectures. Beyond fine-tuning, results with the L1-Sparse method (Liu et al., 2024a; Hayes et al., 2024) show privacy losses once again predict the number of steps to unlearn.

Privacy losses may also offer a more refined measure of data difficulty for unlearning. We show that privacy losses are strongly correlated with some (cheaper-to-compute) proxy measures of data difficulty studied in prior work. On the other hand, we show that privacy losses identify more "difficult" data, which may be useful for evaluation more broadly.

Besides standard empirical unlearning metrics, we evaluate privacy losses on a novel loss-landscape-based metric, based on linear connectivity and loss barriers (Frankle et al., 2020; Fort et al., 2020). While past unlearning metrics look at the output (e.g., loss) of an individual model, loss barriers capture some of the geometry of the loss landscape. Under this stringent test, fine-tuning is able to successfully unlearn. Interestingly, loss barriers also give insight into the differences between difficult and easy data points: points with higher privacy losses start off with larger loss barriers which need to be overcome to unlearn.

Our contributions can be summarized as follows:

- **Per-Instance Theoretical Analysis of Unlearning.** We revise Chien et al.'s analysis of noisy gradient descent for unlearning-via-fine-tuning, replacing a worst-case Rényi-DP bound with recent *per-instance privacy losses* (Thudi et al., 2024). By exploiting that typical

data points may have far less influence on the learned model, our per-instance analysis uncovers an improved trade-off between unlearning and utility compared to prior worst-case analyses.

- **Empirical Validation of Unlearning Difficulty.** We demonstrate the practical applicability of our theoretical insights, showing that per-instance privacy losses reliably predict unlearning difficulty in experiments, both for noisy gradient descent and standard fine-tuning.

- **Efficient Proxies for Privacy Losses.** We demonstrate that cheap proxy measures of data difficulty are strongly correlated with per-instance privacy losses. This identifies practical and scalable alternatives for assessing unlearning quality and forget set difficulty, particularly in resource-constrained settings where full privacy loss computation might be prohibitive.

- **Improved Identification of Difficult Data.** Unlike existing heuristics for capturing aspects of unlearning difficulty, we show that privacy losses identify harder groups of data points, offering a versatile and theoretically justified measure of unlearning difficulty.

- **Loss Barrier Analysis of Unlearning.** We introduce loss barriers (modulo permutation) as a way to evaluate unlearning. We find that the loss barrier is significantly reduced after unlearning, reaching levels comparable to baseline levels.

- **Broader Implications for Unlearning Methodology.** Our empirical findings suggest that per-instance privacy losses may be useful in adapting other fine-tuning-based algorithms, such as L1-sparse.

## 2. Related Work

**Unlearning** Unlearning (Cao & Yang, 2015) aims to remove the influence of training data. In the context of neural networks and non-convex learning problems, even highly optimized exact approaches (Bourtoule et al., 2021) are computationally intensive. Going beyond exact unlearning, Ginart et al. (2019) introduced a notion of *approximate* unlearning, inspired by differential privacy (Dwork et al., 2014). This approach relies on approximate statistical indistinguishability between retraining and unlearning, in exchange for better efficiency or utility compared to exact unlearning. Guo et al. (2020); Neel et al. (2021) studied approximate unlearning algorithms for convex models based on gradient descent using all the training data except those data to be forgotten (a.k.a., *fine-tuning*), followed by the addition of noise (often via the Gaussian mechanism, Dwork et al. 2006) to obtain statistical indistinguishability. Neel et al. (2021) proved that, under various assumptions, their

approach achieves approximate unlearning in a sequential framework with fixed per-deletion runtime.

For non-convex models, a plethora of approximate unlearning methods have been proposed with empirical validation, rather than theoretical guarantees (Graves et al., 2021; Goel et al., 2022; Thudi et al., 2022; Kurmanji et al., 2024; Liu et al., 2024a; Fan et al., 2023). While these approaches are shown to work on some metrics, recent work shows that several unlearning methods fail against more sophisticated attacks for either privacy or poisoning (Hayes et al., 2024; Pawelczyk et al., 2024).

Given the heuristic nature of many unlearning methods, a line of work has attempted to obtain a better understanding of their failure modes. Zhao et al. (2024); Baluta et al. (2024); Fan et al. (2025); Barbulescu & Triantafillou (2024) identified properties of forget sets that influence the behaviors of approximate unlearning algorithms. Zhao et al. (2024); Barbulescu & Triantafillou (2024) also derived improved unlearning methods based on these insights, for vision classifiers and LLMs, respectively. However, we lack a theoretical understanding of the underlying relationship between the identified factors and unlearning difficulty.

Recently, Chien et al. (2024) studied Langevin dynamics for unlearning and introduced an approximate method with privacy guarantees for non-convex models that we build upon in this work, replacing a worst-case bound by a recent per-instance privacy bound. Their method leverages noisy gradient descent, for both the training algorithm, as well as during (noisy) fine-tuning at unlearning time. While our theoretical analysis considers that same setup, we empirically also investigate standard training without noise addition.

**Per-Instance Differential Privacy** Differential privacy (DP) is a standard approach to privacy-preserving data analysis (Dwork et al., 2006; 2014) and machine learning (Chaudhuri et al., 2011; Abadi et al., 2016). DP methodology provides an upper bound on the divergence between output distributions under neighbouring datasets, i.e., when only one individual data point is altered. Since this upper bound needs to hold for every dataset and all of its neighbors, DP is a *worst-case* privacy notion with a single privacy parameter, shared among *all* individual data points. However, in practice, different individual points have different effects when training a machine learning model (Thudi et al., 2024; Yu et al., 2023). For instance, some data points have a much smaller gradient norm than others, making the worst-case privacy analysis for these points over-conservative, and leads to unnecessary degradations in privacy-utility trade-offs. To mitigate such degradations, Ghosh & Roth (2011) and Cummings & Durfee (2020) have analyzed individual sensitivity rather than worst-case sensitivity, which led to an alternative but weaker notion of privacy, called per-instance (or individual, or personalized) DP. Per-instance DP as-

signs a different privacy loss for different data points in a dataset, i.e., it is not worst-case, and it is less pessimistic than standard DP for points that do not affect the output distribution as much as the worst-case points. Building on per-instance DP, Ebadi et al. (2015) and Feldman & Zrnic (2021) have introduced algorithms that filter out data points once their per-instance DP loss exceeds a budget in database query problems and neural network training, respectively. For iterative algorithms like SGD, per-instance DP requires new composition theorems as the privacy loss is adaptive to the individual data points and is different at each iteration (Wang, 2019). Feldman & Zrnic (2021); Thudi et al. (2024) have filled this gap by providing new privacy composition analyses using the associated divergences computed at an individual level.

Motivated by the empirical observation that per-instance privacy loss provides much more promising guarantees than the worst-case DP analysis, we propose to port this per-instance approach to unlearning. We will see that this approach will allow us to provide unlearning guarantees and uncover properties of data that makes a forget set easy to unlearn.

## 3. Preliminaries and Problem Setup

Given a dataset $\mathcal{D} = \{x_i\}_{i=1}^n$ of $n$ points, we are interested in estimating the difficulty of unlearning as a function of the particular *forget set* $\mathcal{D}_F \subset \mathcal{D}$ we are aiming to unlearn.

We are interested in settings where retraining the model from scratch on the *retain set* $\mathcal{D}' = \mathcal{D} \setminus \mathcal{D}_F$ is too costly, and so we consider approximate unlearning. In this work, we adopt a notion of approximate unlearning based on Rényi divergences. For $\alpha > 0$ and $\alpha \neq 1$, recall that the $\alpha$-Rényi divergence of $\mu$ relative to $\nu \gg \mu$, is defined to be

$$D_\alpha(\mu\|\nu) = \frac{1}{\alpha-1} \log\left(\mathbb{E}_{x\sim\nu}\left[\frac{\mu(x)}{\nu(x)}\right]^\alpha\right). \quad (1)$$

(Here, one can interpret $\frac{\mu(x)}{\nu(x)}$ as a Radon–Nikodym derivative more generally.)

For a *fixed* training set and forget set, the following definition captures the goal of unlearning in Rényi divergence:

**Definition 3.1.** Fix a dataset $\mathcal{D}$ and forget set $\mathcal{D}_F \subset \mathcal{D}$. Let $\nu$ be the distribution of the output of $\mathcal{A}(\mathcal{D}')$, i.e., the learning algorithm on the retain set $\mathcal{D}' = \mathcal{D} \setminus \mathcal{D}_F$, and let $\mu$ be the distribution of the output of $\mathcal{U}(\mathcal{A}(\mathcal{D}), \mathcal{D}')$, i.e., the unlearning algorithm on the learned model $\mathcal{A}(\mathcal{D})$ and $\mathcal{D}'$. For $\alpha > 1$, we say $\mathcal{U}$ $(\alpha, \varepsilon)$-*Rényi unlearns* $\mathcal{D}_F$ *from* $\mathcal{A}(\mathcal{D})$ when $D_\alpha(\mu\|\nu) \leq \varepsilon$.

This leads to a *uniform* notion of Rényi unlearning, which offers the same guarantee for all datasets and all forget sets, which is analogous to standard notions.

**Definition 3.2** (Rényi Unlearning). For $\alpha > 1$, an unlearning algorithm $\mathcal{U}$ is an $(\alpha, \varepsilon)$-Rényi unlearner (for $\mathcal{A}$) when, for all dataset $\mathcal{D}$ and forget sets $\mathcal{D}_F \subseteq \mathcal{D}$, $\mathcal{U}$ $(\alpha, \varepsilon)$-Rényi unlearns $\mathcal{D}_F$ from $\mathcal{A}(\mathcal{D})$ in the sense of Definition 3.1.

Note that when this definition holds, one can immediately derive (standard) approximate DP-based unlearning guarantees (Ginart et al., 2019; Guo et al., 2020; Neel et al., 2021; Sekhari et al., 2021), using standard reductions (Proposition 10, Mironov, 2017).[1]

### 3.1. Learning–Unlearning with Noisy Gradient Descent

We consider a learning–unlearning setup based on using projected noisy gradient descent during both learning and unlearning, as studied recently by Chien et al. (2024). In this section, we present theoretical results that rely on noise. In Section 6, we present empirical evidence from multiple benchmarks that the overall trends extend to standard variants of gradient descent.

Formally, our learning algorithm $\mathcal{A} = \mathcal{A}_T$ is $T$ steps of noisy gradient descent on $\mathcal{D}$, starting from a random initialization. Let $\nu_{T,\mathcal{D}}$ denote the distribution on weights obtained after $T$ steps of training on $\mathcal{D}$, and let $\nu_\mathcal{D} = \nu_{\infty,\mathcal{D}}$ denote the stationary distribution, to which the output distribution converges as $T \to \infty$.

Let $\mathcal{D}' = \mathcal{D} \setminus \mathcal{D}_F$ denote the retain set of an unlearning request. Assuming we trained for $T$ iterations, exact unlearning would produce weights whose (marginal) distribution was $\nu_{T,\mathcal{D}'}$, i.e., the distribution as if we had trained on $\mathcal{D}'$ from scratch for $T$ iterations.

Our unlearning algorithm $\mathcal{U} = \mathcal{U}_k$ runs $k$ steps of noisy gradient descent on $\mathcal{D}'$. We denote by $\rho_{\mathcal{D}'}^k(\nu_{T,\mathcal{D}})$ the output distribution of unlearning, i.e., of running $k$ steps of projected noisy gradient descent on $\mathcal{D}'$, initialized at a sample from $\nu_{T,\mathcal{D}}$, i.e., first training on $\mathcal{D}$.

## 4. Per-Instance Unlearning Difficulty Analysis

Our goal is to bound the number of steps, $k$, of unlearning by noisy fine-tuning needed to achieve unlearning in a way that adapts to the influence each data point has on the learned model. We do so by building on two pieces of work: a convergence analysis of Langevin dynamics by Chien et al. (2024) and a per-instance Rényi-differential privacy analysis by Thudi et al. (2024).

---

[1] A symmetrized variation of this definition was proposed by Chien et al. (2024). Our asymmetric version is more consistent with the unlearning literature, where we care about making events under unlearning distribution not much more likely than under the retraining distribution.

### 4.1. Convergence Analysis

Fix a dataset $\mathcal{D}$ and retain set $\mathcal{D}' \subset \mathcal{D}$. We begin with the following corollary of (Thm. 3.2, Chien et al., 2024), which highlights the role of an initial bound on $D_\alpha(\nu_{T,\mathcal{D}} \| \nu_{T,\mathcal{D}'})$. In the following, we assume the loss is Lipschitz continuous and smooth, and that the step sizes used by $\mathcal{A}$ and $\mathcal{U}$ have been set according to (Thm. 3.2, Chien et al., 2024).

**Corollary 4.1.** *Fix $\mathcal{D}$, $\mathcal{D}'$. Let $\{\varepsilon_\alpha, \varepsilon_\alpha'\}_{\alpha \geq 1}$ satisfy $\max\{D_\alpha(\nu_{\mathcal{D}'} \| \nu_{T,\mathcal{D}'}), D_\alpha(\nu_{T,\mathcal{D}'} \| \nu_{\mathcal{D}'})\} \leq \bar{\varepsilon}_\alpha$ and $D_\alpha(\nu_{T,\mathcal{D}} \| \nu_{T,\mathcal{D}'}) \leq \varepsilon_\alpha'$ for all $\alpha$. Then, there exists a constant $C > 0$ such that, for all $\alpha > 1$, $\mathcal{U}_k$ $(\alpha, \epsilon^*)$-Rényi unlearns $\mathcal{D} \setminus \mathcal{D}'$ from $\mathcal{A}(\mathcal{D})$ in $k$ steps, where $\epsilon^*$ is*

$$\left( \frac{2\alpha - 1/2}{2\alpha - 2} \varepsilon_{4\alpha}' + \frac{2\alpha - 1}{2\alpha - 2} \varepsilon_{4\alpha - 1} \right) \exp\left( -\frac{Ck}{2\alpha} \right) + \varepsilon_{2\alpha - 1}.$$

The proof is in Appendix A.1, and follows from (Thm. 3.2, Chien et al., 2024) and the weak triangle inequality for Rényi divergences.

The key observation is that the first term decays exponentially fast in the number of steps $k$, with the initial value determined by the per-instance guarantee and distance to stationarity and the rate determined by the desired value of $\alpha$ and problem parameters (like the Lipschitz and smoothness constants, behind $C$). The second term, $\varepsilon_{2\alpha-1}$, however, does not vanish.

The irreducible term captures the distance to stationarity after $T$ steps of training, measured in a higher order $(2\alpha - 1)$-divergence. This term exists as a consequence of our unlearning algorithm not attempting to correct for the $k$-steps of extra training.

### 4.2. Unlearning Individual Data Points

The following definition is adapted from a differential privacy result by Thudi et al. (2024) to our unlearning setting:

**Definition 4.2** (Per-Instance Privacy Loss). Recall that $\nu_{T,\mathcal{D}}$ is the distribution of the model weights after $T$ iterations of noisy gradient descent on $\mathcal{D}$. Let $g(x^*, w) = \nabla_w \ell(w, x^*)$ be the contribution to the weight-gradient at $w$, coming from one data point $x^*$. The *per-instance privacy loss* for $x^*$ is:

$$P(x, \alpha) := \sum_{t=1}^T C_{t,\alpha} \ln \mathbb{E}_{w \sim \nu_{t,D}} f_{t,\alpha}(\|g(x^*, w)\|_2),$$

where $C_{t,\alpha} = \frac{1}{\alpha - 1} \frac{(p-1)^{t+1}}{p^{t+1}}$ and $\ln f_{t,\alpha}(g)$ is

$$p \ln \left( \sum_{k=0}^{o_p^i(\alpha)} \binom{o_p^t(\alpha)}{k} \mathbb{P}_{x^*}(1)^k \overline{\mathbb{P}_{x^*}(1)}^{o_p^t(\alpha) - k} e^{\frac{g^2(k^2 - k)}{2\sigma^2}} \right),$$

with $o_p(\alpha) = \frac{p}{p-1}\alpha - \frac{1}{p}$ and $o_p^t$ is $o_p$ composed $t$ times, $\mathbb{P}_{x^*}(1) = 1 - \overline{\mathbb{P}_{x^*}(1)}$ is the sampling probability of the data point (batch size over dataset size), and $p$ is a free parameter we set to $3T$, following (Fact 3.4, Thudi et al., 2024).

This quantity, which we show how to estimate efficiently in Section 5.1, yields a bound on a data point's sensitivity.

**Theorem 4.3.** *Fix $\mathcal{D}$, let $x \in \mathcal{D}$, and put $\mathcal{D}' = \mathcal{D} \setminus \{x\}$. Then $D_\alpha(\nu_{T,\mathcal{D}} \| \nu_{T,\mathcal{D}'}) \leq P(x, \alpha)$.*

*Proof.* This follows by applying the per-instance moment-based composition theorem (Thm. 3.3, Thudi et al., 2024) with the per-step divergence bound for single data points (Thm. 3.2, Thudi et al., 2024), using the post-processing inequality to conclude that the divergence after projections is bounded by that before projections. □

As a consequence of Theorem 4.3 applied to Corollary 4.1, we have unlearning individual data points by noisy gradient descent depends logarithmically on the privacy loss. The following corollary is a direct substitution of Theorem 4.3 into Corollary 4.1.

**Corollary 4.4.** *Under the assumptions of Corollary 4.1 and Theorem 4.3, for all $\mathcal{D}$ and $\mathcal{D}' = \mathcal{D} \setminus \{x\}$ for $x \in \mathcal{D}$, for all $\alpha > 1$, there exist constants $A_\alpha, B_\alpha, C_\alpha > 0$ such that, for all $\delta > \varepsilon_{2\alpha-1}$, running noisy gradient descent for*

$$k \geq A_\alpha \ln \left( \frac{B_\alpha P(x, 4\alpha) + C_\alpha \varepsilon_{4\alpha-1}}{\delta - \varepsilon_{2\alpha-1}} \right)$$

*steps $(\alpha, \delta)$-unlearns $\{x\}$ from $\mathcal{A}(\mathcal{D})$.*

### 4.3. Limitations of Existing Group Unlearning Analyses

The above analysis focuses on forgetting one data point. How about bounding the work to unlearn multiple data points simultaneously? The group unlearning bound of (Chien et al., 2024, Cor. 3.4) requires that the order of $\alpha$ grows with each data point to unlearn. This is problematic, as the Rényi divergence between, e.g., Gaussians, grows linearly with $\alpha$ (Prop. 7, Mironov, 2017). In contrast, (Thms. 3.3 and 3.6, Thudi et al., 2024) imply that the Rényi divergence $D_\alpha(\nu_{T,\mathcal{D}} \| \nu_{T,\mathcal{D} \setminus \mathcal{D}_F})$, and hence steps to unlearn, does not necessarily grow with $\mathcal{D}_F$; instead, Thudi et al. bound the divergence by comparing the distribution of gradients under $\mathcal{D}$ and under $\mathcal{D} \setminus \mathcal{D}_F$. Implementing their group privacy accounting (described in Appendix B), however, we found evidence the bounds were likely too loose, as they did not differentiate forget sets that we empirically knew to require different numbers of steps to unlearn (see Figure 4 in appendix). In contrast, we found the rankings provided by privacy losses meaningfully differentiate the number of steps needed to unlearn, as we will describe in Section 6.

We conclude that the current state of analysis for group unlearning is not tight enough to capture the behavior we observe in practice. The problem of obtaining a tight analysis remains an open problem.

## 5. Methodology

Our empirical methodology is structured around two key objectives driven by the theoretical role of per-instance privacy losses in unlearning. First, we aim to validate that privacy losses effectively predict the relative difficulty of unlearning data points. Second, we seek to understand the factors contributing to this difficulty by investigating the relationship between privacy losses, the loss landscape, and existing metrics of data difficulty. In the following sections, we present our experimental design to address these questions.

### 5.1. Empirically Validating Unlearning Difficulty

**Unlearning Algorithms** Our investigation primarily focuses on unlearning via fine-tuning on the retain set. We also run unlearning experiments with L1-Sparse (Liu et al., 2024a), a regularized version of fine-tuning that is widely recognized as one of the most effective unlearning methods. Hayes et al. (2024) has demonstrated that L1-Sparse outperforms a number of other methods in defending against a basic Membership Inference Attack (MIA), as well as other attacks of varying strengths.

**Per-instance Privacy loss** We compute the terms in the privacy loss $P(x, \alpha)$ stated in Definition 4.2 by taking a Monte-Carlo estimate from a single training run with checkpoints $w_0, w_{s_1}, w_{s_2}, \cdots, w_{s_N}$, i.e.,

$$\hat{P}_{s_i}(x, \alpha) = C_{s_i, \alpha} \ln f_{s_i, \alpha}(\|g(x^*, w_{s_i})\|_2),$$

where $C_{s_i, \alpha}, f_{s_i, \alpha}, g$ are defined in Definition 4.2.

We then approximate the area under the per-step privacy curve (i.e., sum over $t = 0, 1, \cdots, s_N$) by using the right hand rule: keeping $\hat{P}_{s_i}$ constant between the checkpointing intervals $(s_{i-1}, s_i]$. This gives our approximate privacy loss:

$$\hat{P}(x, \alpha) = \sum_{i=1}^N \hat{P}_{s_i}(x)(s_i - s_{i-1}).$$

Throughout the paper we take $N = 35$ and have the checkpoints $s_i$ evenly spaced throughout training. In the case of SGD, without explicit noise, we approximate these scores by assuming a negligible amount of noise is present. In particular we take $\sigma \leq 0.1$, which matches the trends we observe with SGLD. We also found the privacy losses rankings for SGD to be stable to the implicit $\sigma$ used for privacy losses computation (see Appendix F). We note that past work has looked into quantifying the noise inherent in training, due to hardware and software nondeterminism (Jia et al., 2021; Zhuang et al., 2022). It is an open problem to exploit software and hardware nondeterminism to offer a theoretical justification to our approach here.

**Forget Set Difficulty** We rank examples by privacy losses and form 5 forget sets of varying difficulty by taking evenly

spaced sequences of 1000 data points. The size of the forget set was chosen to be in a similar range as prior work (e.g., (Zhao et al., 2024)). Our theory suggests that higher privacy losses should lead to longer, thus more difficult, unlearning. We thus call a forget set more difficult than another forget set if its average privacy loss is higher. In our analysis, each forget set is represented by the average privacy loss of all samples within that set. Further methodological details are provided in Appendix D.

**Unlearning Evaluation**   We evaluate unlearning efficacy using three metrics: (1) accuracy, measured separately on the retain set (RA), test set (utility), and on the forget set (FA). We report $\text{UA} = 1 - \text{FA}$, indicating how "accurate" unlearning is, as done in (Fan et al., 2023); (2) membership inference attack (MIA): we train a logistic regression classifier to identify training samples and report the fraction of forget set samples incorrectly classified as test samples, thus indicating successful forgetting; and (3) Gaussian Unlearning Score (GUS) (Pawelczyk et al., 2024), which employs Gaussian input poisoning attacks to reveal if the unlearned model still encodes noise patterns associated with the poisoned forget set data. See Appendix E for more details.

These metrics are monitored during unlearning and compared with the oracle model to evaluate the effectiveness of the unlearning methods. Recall that the oracle model is obtained by training from scratch using the retain set only. We choose these metrics due to their common use in machine unlearning research (Fan et al., 2025; Zhao et al., 2024; Jia et al., 2023; Deeb & Roger, 2024; Fan et al., 2024; Pawelczyk et al., 2024) and their computational simplicity, enabling us to compute them at every step during unlearning. Additional details on their computation and associated parameter choices can be found in Appendix E.

**Datasets and models**   Our experiments are performed on the SVHN (Netzer et al., 2011) and CIFAR-10 (Alex, 2009) datasets, with a ResNet-18 architecture (He et al., 2016). Appendix F presents results for ViT-small (Dosovitskiy, 2020). Appendix D describes other details for reproducibility.

Each experimental configuration (oracle training, or training on all data and unlearning) is run 10 times, and the average performance across these runs is reported.

### 5.2. Loss Landscape Analysis

While per-instance privacy losses provide a quantitative measure of unlearning difficulty based on training dynamics, they do not directly reveal the underlying geometric properties of the loss landscape that contribute to this difficulty. To gain a deeper understanding of why certain data points are harder to unlearn, we complement our privacy loss analysis with an investigation of the loss landscape.

Specifically, we employ the concepts of (linear) loss barriers (Frankle et al., 2020; Nagarajan & Kolter, 2019). Loss barriers characterize the "flatness" or "curvature" of the loss surface between different model parameter configurations.

The loss barrier $\text{err}(w, w'; \mathcal{D})$ is the deviation in cross entropy $\mathcal{L}$ on the data $\mathcal{D}$ along the linear path in weight space connecting $w$ to $w'$. Let $\bar{\alpha} = 1 - \alpha$. Then $\text{err}(w, w'; S)$ is

$$\max_{\alpha \in [0,1]} \left[ \mathcal{L}(\alpha w + \bar{\alpha} w'; S) - \alpha \mathcal{L}(w; S) - \bar{\alpha} \mathcal{L}(w'; S) \right]. \quad (2)$$

To account for the permutation invariance of neural networks, we compute these loss barriers modulo permutation, as detailed in (Entezari et al., 2022; Sharma et al., 2024).

Loss barriers provide insight into the geometric properties of high-dimensional loss surfaces. In our experiments, we compute loss barriers between oracle models (trained without the forget set) and models before and after unlearning forget sets with various average privacy losses. This allows us to examine if forget sets with higher privacy losses (higher predicted unlearning difficulty) exhibit distinct loss landscape characteristics, particularly in terms of loss barriers.

### 5.3. Comparison to Alternative Data Difficulty Metrics

While the per-instance privacy losses provide valuable insights into the unlearning process, their computation requires storing gradients throughout training, leading to considerable computational overhead. Therefore, we explore alternative metrics that could serve as proxies for these scores, offering a more efficient way to estimate forget set difficulty. Furthermore, we investigate the relationship between these proxies and fine-tuning-based unlearning difficulty, shedding light on their underlying mechanisms.

We evaluate five proxies: (1) the gradient norm of individual data points at a single mid-training iteration; (2) the gradient norm at the end of training; (3) the average gradient norm across all training iterations; (4) C-Proxy used by Zhao et al. (2024) to approximate memorization scores from (Feldman et al., 2018); adapted from Jiang et al. (2020). This proxy computes prediction confidence: the entry in the softmax vector corresponding to the ground truth class, averaged throughout the training trajectory; and (5) a single-trajectory variant of the EL2N score (Paul et al., 2021). Normally, EL2N score is computed by averaging error signals over multiple training trajectories at a fixed time point. However, for computational efficiency and direct comparability with our gradient-based proxies, we compute a single-trajectory EL2N score at a mid-training checkpoint. See Appendix C for a discussion on connections among the scores.

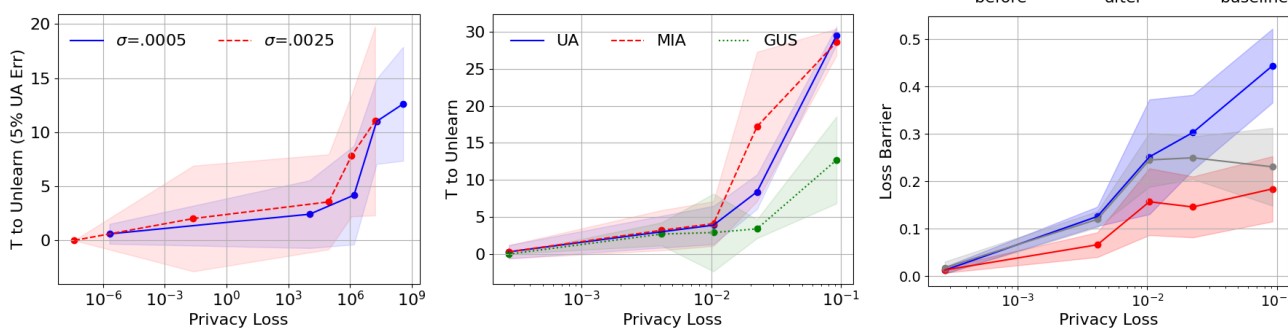

*Figure 1.* CIFAR-10 dataset results. **Left:** SGLD unlearning with varying levels of noise ($\sigma$). Forget set difficulty (x-axis), as measured by the privacy loss, against time to unlearn (y-axis). Time to unlearn is measured in terms of epochs needed to get within 5% of the unlearning metric (e.g., UA or MIA) measured on the oracle model. **Middle:** SGD unlearning. Time to unlearn measured across three evaluation metrics. **Right:** Error barrier between the oracle and the unlearned model before and after unlearning for forget sets with different privacy losses. Baseline corresponds to the loss barrier between two oracles.

## 6. Experimental Results

We empirically validate the predictive capabilities of per-instance privacy losses in forecasting unlearning difficulty. Our primary finding is that privacy losses accurately rank datapoints according to the number of steps needed to unlearn. This is tested for two settings of interest: (1) SGLD, which aligns directly with the assumptions of our theoretical upper bounds; and (2) SGD. For SGD, while lacking explicit noise, we adapt privacy losses by assuming a small, implicit noise component.

To probe the geometric origins of unlearning difficulty as captured by privacy losses, we analyze the loss landscape. Our investigation reveals a consistent trend: data points with higher privacy losses exhibit larger loss barriers on a linear path (in the weight space) to the oracle model.

We also assess the practical utility of privacy losses by comparing them to computationally efficient proxy metrics. Our results demonstrate a strong correlation between privacy losses and existing proxy metrics, including those employed in previous studies to estimate forget set difficulty (C-Proxy in (Zhao et al., 2024)). However, a key advantage emerges: privacy losses provide superior precision in identifying truly difficult-to-unlearn data points within the context of fine-tuning-based unlearning. The forget sets ranked as most difficult by privacy losses consistently take longer to unlearn than those identified by these established proxies.

### 6.1. Time to Unlearn Depends on Per-Instance Privacy

We find that across a variety of unlearning metrics, privacy losses accurately separate data points by the number of steps needed to unlearn in both SGLD and SGD. Appendix F.2 shows the same trends for L1-sparse unlearning.

**SGLD Training** To evaluate the predictions outlined in Section 4, we examine the relationship between the number of steps needed to unlearn using noisy fine-tuning (SGLD), and the average privacy loss within the forget set. Figure 1 (left) shows that as forget set privacy losses increase, a higher number of fine-tuning steps is needed to reach a 5% error margin relative to the oracle, validating our prediction.

**SGD Training** While SGLD offers theoretical advantages for privacy analysis and bounding Rényi unlearning, SGD remains the dominant training paradigm in practice. Therefore, we investigate whether our theoretical framework, developed in the context of Langevin dynamics, can also predict unlearning time for models trained with SGD.

As shown in Figure 1 (middle), we observe a similar trend as in the SGLD experiments across all evaluation metrics: unlearning more difficult forget sets, characterized by higher average privacy losses, requires more fine-tuning steps.

This suggests that even without explicit noise injection during training, the concept of per-instance privacy losses derived from our theoretical analysis can provide valuable insights into the unlearning process for SGD-trained models. That is, SGD seems to be well-approximated by low noise SGLD. These results are replicated across additional datasets, architectures, and forget set sizes, with full details and qualitative examples in Appendix F.

### 6.2. More Difficult Forget Sets, Larger Loss Barriers

We characterize difficult to unlearn data points by their loss landscape, in particular, loss barriers (see Section 5).

Figure 1 (right) depicts the loss barrier between oracle models and unlearned models, while varying the difficulty of the forget set, as measured by the average privacy loss.

We observe two key takeaways from this analysis: **(1)** Comparing our results to the baseline loss barriers between independently trained oracle models, we find that fine-tuning achieves comparable levels after unlearning. This can be seen as a measure of unlearning efficacy based on loss barriers: if the (distribution of the) loss barrier is different from that of the baseline between oracles, unlearning was unsuccessful; **(2)** Higher privacy loss values correspond to larger initial barriers, providing a geometric interpretation for privacy losses; data points that require more steps to unlearn have to overcome larger loss barriers to the oracle.

Overall, our findings provide additional evidence for the utility of privacy losses in predicting unlearning difficulty, and point to the effectiveness of loss barriers as both a diagnostic and an evaluation tool for machine unlearning.

### 6.3. Existing Metrics Correlate with Privacy Losses

We now investigate how privacy losses compare to other data difficulty metrics, as described in Section 5. Our results in Figure 2 reveal that all these proxies exhibit high correlation with the actual privacy loss. As expected, the best proxy is the average gradient norm throughout training, but it is also the most expensive proxy as it requires computing gradient norms for each data point at every iteration, versus once. Other proxies (with the exception of C-Proxy) only require a gradient/error computation at a single checkpoint, yet still provide a reasonable approximation for categorizing examples into broad difficulty groups.

These findings suggest that, in scenarios where computational resources are limited, utilizing these proxies can offer a practical alternative for estimating forget set difficulty and predicting unlearning time. We refer to recent work by Kwok et al. (2024) for an in-depth comparison of different data difficulty metrics that may serve as good proxies.

Recall that C-Proxy has been used in prior work by Zhao et al. (2024) to identify difficult to unlearn forget sets for certain unlearning algorithms. Figure 2 shows that privacy losses are highly correlated with this heuristic. Our work thus offers theoretical grounding to previously proposed heuristics for identifying difficult to unlearn forget sets.

Finally, note that among metrics relying on averaging over training, our method is more efficient, requiring only 35 evenly spaced checkpoints (approximately 20% of training time), compared to C-Proxy and average gradient norm, which store values at every epoch (150 in total). As shown in Section 6.4, it is also more effective at identifying hard-to-unlearn samples than all other metrics, regardless of whether they rely on averaging.

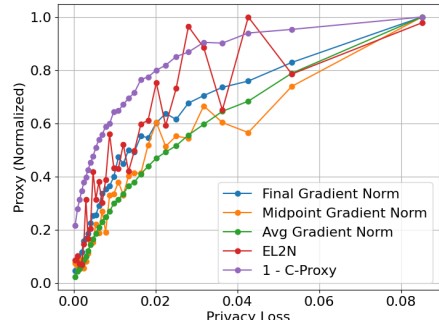

*Figure 2.* Correlation between privacy losses (x-axis) and various proxy metrics (y-axis). The values of all proxy metrics are normalized to their maximum value for better visual clarity. For improved readability, the data is binned into 30 bins.

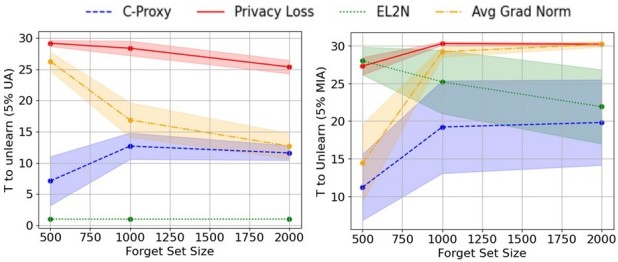

*Figure 3.* Comparison of the time needed to unlearn (y-axis) the most difficult forget sets as identified by privacy losses (ours), C-Proxy, average gradient norm and EL2N, across different forget set sizes (x-axis).

### 6.4. Privacy Losses Identify Harder Data

In the previous subsection, we provided evidence that existing data difficulty metrics correlate with per-instance privacy losses. In this section, we provide evidence that, in fact, privacy losses are able to identify harder data. In Figure 3, we compare our per-instance privacy losses to several data difficulty metrics, including C-Proxy (described in Section 5), average gradient norm, and EL2N scores.

For forget sets of size $s \in \{600, 1000, 2000\}$, we look at the top $s$ data points as ranked by C-Proxy, average gradient norm, EL2N scores, and privacy losses. What we see is that, across the board, the data picked out by privacy losses are harder to unlearn, as measured both by the number of iterations to reach 5% unlearning accuracy or reach 5% excess membership inference attack error. By identifying more difficult examples, privacy losses open up new empirical approaches to evaluating unlearning performance.

## 7. Conclusion

In this work, we have introduced a principled approach to quantifying machine unlearning difficulty at the level of individual data points in terms of per-instance privacy losses,

which bound the Rényi divergence between training with and without a datapoint. Our theoretical analysis provides a foundation for understanding how unlearning scales with the properties of specific data points, particularly in the context of Langevin dynamics.

We have shown that per-instance privacy losses, estimated from training statistics, reliably predict unlearning difficulty in fine-tuning-based unlearning algorithms, across different architectures and datasets, even in the absence of explicit noise injection during training. Our empirical results demonstrate that these privacy losses offer a precise and actionable measure of unlearning difficulty. Our work also offers a theoretical grounding for previous work suggesting that certain forget sets are harder to unlearn, with privacy losses capturing similar aspects of difficulty for fine-tuning-based unlearning as previously proposed heuristics (Zhao & Triantafillou, 2024; Baluta et al., 2024; Zhao et al., 2024). Moreover, we show privacy losses identify harder forget sets than previous methods.

Our findings have broader implications for unlearning methodology, suggesting that per-instance divergence analysis can guide the development of new, more efficient unlearning algorithms tailored to specific data characteristics. Extending our theoretical framework to other unlearning methods beyond fine-tuning and exploring the use of privacy losses in designing adaptive unlearning strategies is a promising direction for future work.

## Impact Statement

This paper presents work whose goal is to advance the field of machine unlearning, which is specifically oriented to improve the trustworthiness of machine learning, by supporting requests to remove the influence of training data. There are many potential positive societal consequences of our work, none which we feel must be specifically highlighted here.

## Acknowledgments

We thank Ioannis Mitliagkas and Ilia Shumailov for feedback on a draft of this work. Anvith Thudi, Ashmita Bhattacharyya, and Nicolas Papernot would like to acknowledge the sponsors of the CleverHans lab, who support our research with financial and in-kind contributions: CIFAR through the Canada CIFAR AI Chair, NSERC through the Discovery Grant, the Ontario Early Researcher Award, the Schmidt Sciences foundation through the AI2050 Early Career Fellow program, and the Sloan Foundation. Resources used in preparing this research were provided, in part, by the Province of Ontario, the Government of Canada through CIFAR, and companies sponsoring the Vector Institute. Anvith Thudi is also supported by a Vanier Fellowship from NSERC. Daniel M. Roy is supported by the funding through NSERC Discovery Grant and Canada CIFAR AI Chair at the Vector Institute. We also thank Mila – Quebec AI Institute and Google DeepMind for providing the computational resources that supported this work.

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

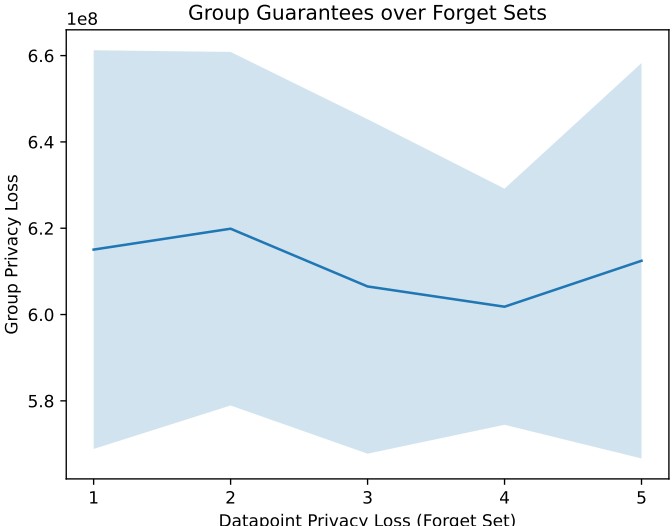

*Figure 4.* We compared our estimates of the group privacy guarantees (y-axis) across forget sets determined by rankings of privacy losses (x-axis), and found the group privacy guarantees did not change. This was despite these forget sets leading to consistent differences in the number of steps to unlearn. We report the mean over 20 estimates of the group privacy values, and one standard deviation. We conclude the theory for group unlearning is currently not sharp enough to capture trends seen in practice.

## A. Proofs

### A.1. Corollary 4.1

*Proof.* By (Theorem 3.2, Chien et al., 2024), for all $\alpha > 1$, there exists a constant $C > 0$ such that

$$D_{2\alpha}(\rho_{\mathcal{D}'}^k(\nu_{T,\mathcal{D}})\|\nu_{\mathcal{D}'}) \leq e^{-\frac{Ck}{2\alpha}} D_{2\alpha}(\nu_{T,\mathcal{D}}\|\nu_{\mathcal{D}'}), \tag{3}$$

where $D_{2\alpha}(\nu_{T,\mathcal{D}}\|v_{\mathcal{D}'})$ is the initial $2\alpha$-Rényi divergence to stationarity on $\mathcal{D}'$, after training on $\mathcal{D}$. By the weak triangle inequality (Proposition 11, Mironov, 2017), this is bounded by

$$\frac{2\alpha - 1/2}{2\alpha - 1}D_{4\alpha}(\nu_{T,\mathcal{D}}\|\nu_{T,\mathcal{D}'}) + D_{4\alpha-1}(\nu_{T,\mathcal{D}'}\|\nu_{\mathcal{D}'}) \leq \frac{2\alpha - 1/2}{2\alpha - 1}\varepsilon'_{4\alpha} + \varepsilon_{4\alpha-1}, \tag{4}$$

where the second inequality follows from our hypotheses.

Finally, applying the weak triangle inequality once more, $D_\alpha(\rho_{\mathcal{D}'}^k(\nu_{T,\mathcal{D}})\|\nu_{T,\mathcal{D}'})$ is bounded by

$$\frac{\alpha - 1/2}{\alpha - 1}D_{2\alpha}(\rho_{\mathcal{D}'}^k(\nu_{T,\mathcal{D}})\|\nu_{\mathcal{D}'}) + D_{2\alpha-1}(\nu_{\mathcal{D}'}\|\nu_{T,\mathcal{D}'}),$$

which yields the claimed bound from our hypotheses, after substituting Equations (3) and (4). □

## B. Group Privacy Analysis and Methodology

The following theorem comes from the results of Thudi et al. (2024).

**Theorem B.1.** *Suppose we train with noisy gradient descent for $T$ steps. Then for an arbitrary $\mathcal{D}, \mathcal{D}' = \mathcal{D} \setminus \mathcal{D}_F$, we have:*

$$D_\alpha(\nu_{T,\mathcal{D}}\|\nu_{T,\mathcal{D}'}) \leq \sum_{t=1}^{T} C_{t,\alpha} \ln \mathbb{E}_{w\sim\nu_{t,\mathcal{D}}} G_{t,\alpha}(\mathcal{D}, \mathcal{D}', w).$$

*where $C_{t,\alpha} = \frac{1}{\alpha-1}\frac{(p-1)^{t+1}}{p^{t+1}}$ and*

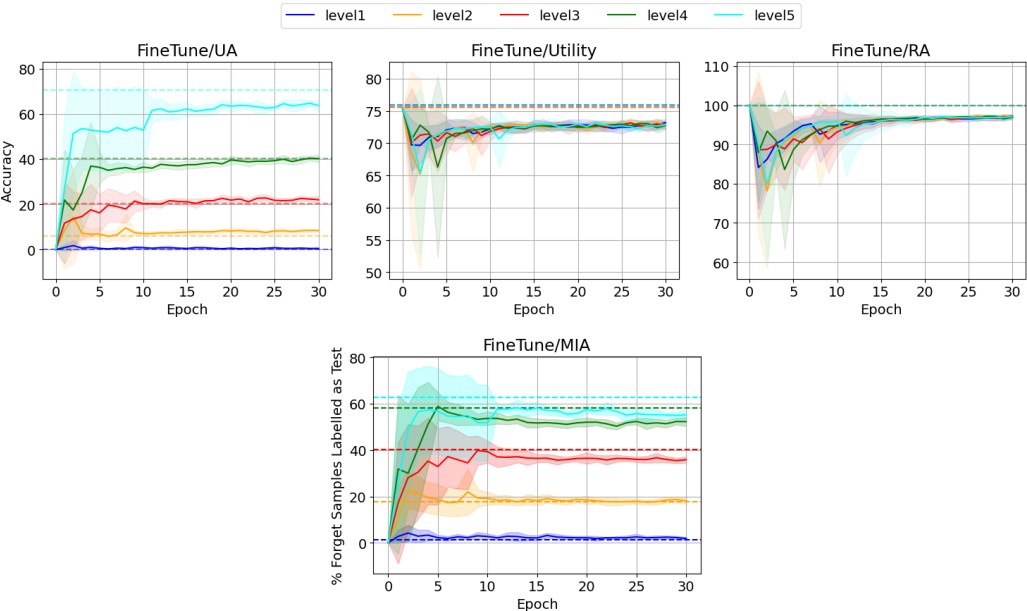

*Figure 5.* Unlearning results for accuracy metrics (top) and MIA success rate (bottom). The x-axis represents the number of epochs. In each plot, lines of different colors represent forget sets of varying difficulty, while the dashed line indicates the oracle's performance.

$$\ln G_{t,\alpha}(\mathcal{D},\mathcal{D}',w) = p\mathbb{E}_{\mathcal{D}'_B} \ln \mathbb{E}_{\mathcal{D}_{\mathbf{B}}\mathbf{o_P^t}(\alpha)} \left( e^{\frac{-1}{2\sigma^2}\Delta_{o_p^t(\alpha)}(\mathcal{D}_{\mathbf{B}}\mathbf{o_P^t}(\alpha),\mathcal{D}'_B,w)} \right)$$

*where $\mathcal{D}_{\mathbf{B}}^\alpha = \{\mathcal{D}_B^1,\cdots,\mathcal{D}_B^\alpha\}$ is a random sample of $\alpha$ minibatches from $\mathcal{D}$, $\mathcal{D}'_B$ is a single random minibatch, $o_p(\alpha) = \frac{p}{p-1}\alpha - \frac{1}{p}$ and $o_p^t$ is $o_p$ composed $t$ times, and $p$ is a free parameter we set to $3T$ following (Fact 3.4, Thudi et al., 2024), and letting $U(\mathcal{D}_B,w) = \nabla_w \ell(w,\mathcal{D}_B)$ be the mini-batch gradient:*

$$\Delta_\alpha(\mathcal{D}_{\mathbf{B}}^\alpha,\mathcal{D}'_B,w) \coloneqq \sum_i \|U(\mathcal{D}_B^i,w)\|_2^2 - (\alpha-1)\|U(\mathcal{D}'_B,w)\|_2^2 - \|\sum_i U(\mathcal{D}_B^i,w) - (\alpha-1)U(\mathcal{D}'_B,w)\|_2^2.$$

*Proof.* A direct consequence of applying (Theorem 3.3, Thudi et al., 2024) with the general update per-step divergence bound of (Theorem 3.6, Thudi et al., 2024), and noting the divergence after applying the projections is bounded by the divergence before applying the projections by the post-processing inequality. □

### B.1. Methodology for privacy losses computation

To estimate the guarantees we take a Monte-Carlo sample of checkpoints from a single training run at steps $s_0, s_1, \cdots, s_N$, and estimate the $\ln G_{t,\alpha}(\mathcal{D},\mathcal{D}',w)$ term at each step by sampling a single random mini-batch from $\mathcal{D}'$ and $o_p^{s_i}(\alpha)$ mini-batches from $\mathcal{D}$ to estimate the expectations. We then compute the sum using the right hand rule, analogous to our estimate of privacy losses described in Section 5.

In Figure 4 we took checkpoints from an SGD training run on CIFAR10 with ResNet18, and used $\sigma = 0.1$, and took $\alpha = 8$ to compute the group privacy scores. We report the mean over 20 estimates (given the stochasticity in our estimates for the per-step terms $\ln G_{t,\alpha}(\mathcal{D},\mathcal{D}',w)$) and shaded in one standard deviation.

## C. Example Difficulty Related Work

Paul et al. (2021) propose EL2N score to capture how much an example contributes to learning a high accuracy predictor, high score meaning high importance during training to achieve high accuracy. At the same time, the authors find that high

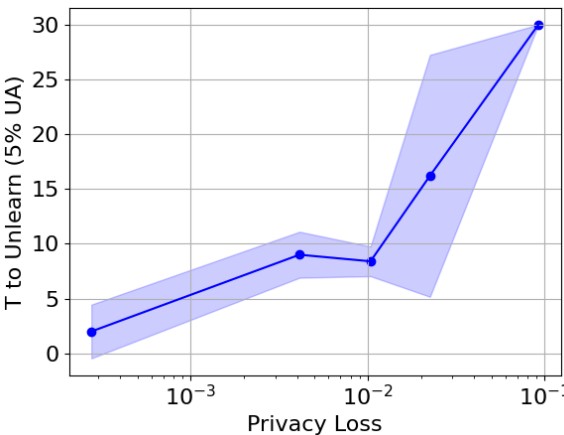

*Figure 6.* Unlearning time for forget sets with different privacy loss values using the L1-sparse method. Time is measured by the number of steps required for the unlearning method to reach a 5% margin of error, where error is defined as the difference between the unlearned model's UA and the oracle's UA for the given forget set.

scoring examples tend to be difficult to learn, are often memorized at the end of training and are outliers. This score has been shown to correlate with a number of other example difficulty and memorization metrics proposed in the literature (Kwok et al., 2024; Paul et al., 2021), some of which have been shown to also capture unlearning difficulty for a number of unlearning algorithms (Zhao et al., 2024; Baluta et al., 2024; Zhao & Triantafillou, 2024).

## D. Additional Experimental Details

**Constructing Difficulty-Based Forget Sets**   To create forget sets with varying difficulty levels, the training dataset is partitioned into five subsets based on privacy scores. First, the samples are sorted in ascending order by their scores. Recursive splits are then performed to identify key thresholds: the lower quartile (Q1), the median (Q2), and the upper quartile (Q3). Using these thresholds, five subsets are constructed: (1) the first 1000 samples, (2) intervals centered around Q1 (Q1 ± 500 samples), (3) intervals centered around Q2 (Q2 ± 500 samples), (4) intervals centered around Q3 (Q3 ± 500 samples), and (5) the last 1000 samples. This approach provides a systematic stratification of the dataset, enabling the evaluation of unlearning performance across varying levels of difficulty as determined by privacy scores.

**Learning rates and training times for SGD**   The original model, which serves as the starting point for all unlearning techniques (not for SGLD), is trained for 150 epochs using an initial learning rate of 0.01, a weight decay of 0.0005, and a learning rate schedule that reduces the learning rate by an order of magnitude at epochs 80 and 120. Each unlearning method is subsequently fine-tuned for 25 epochs.

**Additional details for SGLD**   At every step we added $N(0, \sigma^2)$ Gaussian noise to the minibatch gradient, where we vary $\sigma$ for ablations. All other hyperparameters were kept the same as SGD. In particular we do not do any additional gradient clipping.

**Additional details for L1-sparse**   L1-sparse is an unlearning method inspired by the observation that pruning aids unlearning (Liu et al., 2024a). Its objective function closely resembles that of fine-tuning but includes an additional $L_1$ regularization term, weighted by a hyperparameter $\alpha$, which encourages sparsity in model parameters to facilitate unlearning.

**Hyperparameter tuning**   We perform hyperparameter tuning (HPT) for the unlearning methods using the Bayesian optimization method on a random forget set. While fine-tuning involves a single hyperparameter–the learning rate– L1-Sparse additionally optimizes $\alpha$. To determine the best hyperparameters for each method, we employ Bayesian optimization to find configurations that achieve an optimal balance between privacy and utility. Additionally, to ensure that the selected hyperparameters are also optimized with respect to the number of steps required for unlearning, we identify hyperparameter sets that fall within a 5% margin of error for this trade-off. Among these, we select the configuration that converges the fastest.

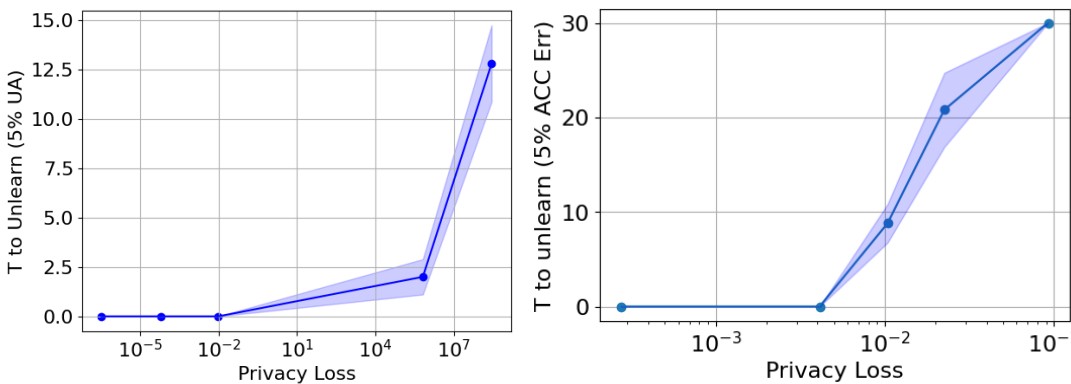

*Figure 7.* SGD unlearning. Unlearning time vs. privacy score for ResNet-18 on SVHN (left) and ViT-small on CIFAR-10 (right). Unlearning time is measured in steps required to reach a 5% UA error margin.

**Compute resources**  Experiments were conducted using L40 and RTX8000 GPUs, and AMD EPYC 7452 CPUs.

## E. Evaluation metrics

**Membership Inference Attack**  Membership inference attacks (MIA) aim to determine whether a given sample was part of a model's training data by analyzing differences in the model's responses. In our approach, we train a logistic regression classifier using the model's confidence values on training and test samples as inputs. The attacker then attempts to classify the forget samples, with success measured by the percentage of forget samples labeled as test, indicating effective unlearning.

**Gaussian Unlearning Score**  We use the Gaussian Unlearning Score (GUS), introduced by Pawelczyk et al. (2024), to quantify the impact of poisoned samples on the model. To compute GUS, each sample in the forget set is perturbed with zero-mean Gaussian noise with a standard deviation of $\sigma$. GUS is then computed by averaging (over the forget set) the per-example inner product between the gradient of the loss with respect to the clean (non-poisoned) sample and the stored Gaussian noise used for poisoning, normalized by the L2 norm of the gradient. The effectiveness of an unlearning method is then assessed by how well it mitigates the influence of these poisoned samples. Specifically, the change in GUS before and after unlearning serves as a measure of unlearning success.

For the CIFAR-10 and ResNet-18 setup, the original work recommends a variance value of $0.32$. However, in our experiments, we explored different values and found that smaller variances were better suited for our setup. Based on these empirical findings, we set $\sigma^2 = 0.062$.

## F. Additional Experimental Results

### F.1. Unlearning trends during fine-tuning

In this experiment, we apply the fine-tuning method to unlearn a ResNet-18 model trained on the full CIFAR-10 dataset. The forget set varies in difficulty, with five different levels determined by the proxy losses of the samples. The UA, utility and RA values during unlearning is depicted in Figure 5 (top), while the MIA results are depicted in Figure 5 (bottom). For UA and MIA, we see the most difficult forget sets do indeed take longer to unlearn. We see utility and RA are similar across difficulty levels.

### F.2. L1-sparse fine-tuning

The unlearning results for the L1-sparse method are presented in Figure 6. Similar to our observations for unlearning with SGD or SGLD fine-tuning, unlearning with L1-sparse takes longer to forget samples with higher privacy scores. In fact, the more challenging the forget sets, the longer the unlearning.

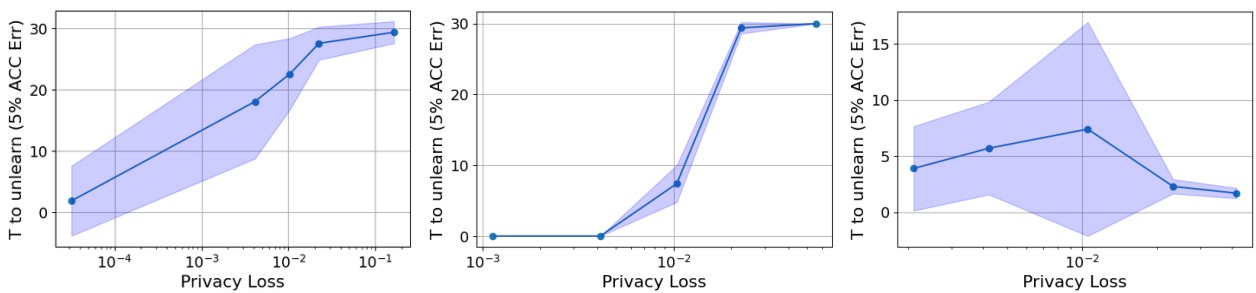

*Figure 8.* Unlearning time for forget sets of size 100 (left), 5,000 (middle), and 10,000 (right), evaluated on CIFAR-10. Unlearning time is reported as the number of training steps required to achieve a UA error within 5%.

### F.3. Additional datasets/architectures

In addition to ResNet-18 on CIFAR-10, we conduct experiments with additional dataset-architecture pairs. Specifically, we evaluate ResNet-18 on SVHN and ViT-small on CIFAR-10. ViT-small is a Vision Transformer model that applies self-attention mechanisms to sequences of image patches, enabling effective global feature extraction. Figure 7 presents the unlearning results for ResNet-18 on SVHN (left) and ViT-small on CIFAR-10 (right). The results suggest that, consistent with our previous findings, unlearning takes longer for forget sets with a higher average privacy loss.

**Rankings across noise levels** We ran experiments to test sensitivity of the ranking for SGD to the noise values used in estimating privacy losses. For SVHN with ResNet-18 we found: (1) Spearman correlation between rankings at $\sigma = 0.01$ and $\sigma = 0.001$ was $0.70(p = 0.0)$. (2) Between $\sigma = 0.001$ and $\sigma = 0.0005$ was $0.99(p = 0.0)$. (3) Between $\sigma = 0.0005$ and $\sigma = 0.0001$ was $0.99(p = 0.0)$. These results suggest that rankings are largely noise-invariant, as long as some noise is present. Past work has shown evidence of observable noise during training due to software and hardware non-determinism (Jia et al., 2021).

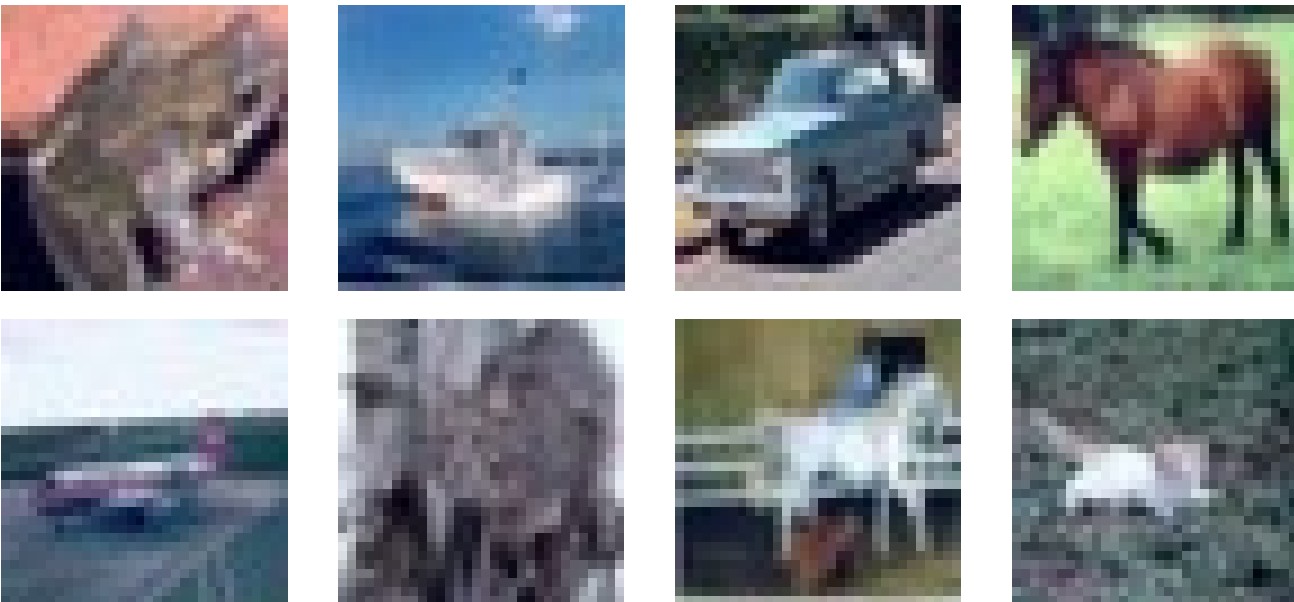

*Figure 9.* Qualitative examples of forgetting difficulty on CIFAR-10. We show 4 examples each from the easy-to-forget (top row) and hard-to-forget (bottom row) subsets, identified from a forget set of 1,000 samples.

## F.4. Varying Forget Set Sizes

We conducted additional experiments varying the size of the forget set (100, 5,000, and 10,000 samples). The results are presented in Figure 8. Our findings indicate that privacy loss consistently distinguishes between easy and hard-to-forget subsets, even when the forget set is small. Interestingly, for very large forget sets (e.g., 10,000 samples), the separation begins to diminish. This observation is intuitive, as the variance in average privacy loss decreases with increasing subset size.

## F.5. Qualitative Results

Qualitative examples of easy and hard to forget CIFAR-10 samples are provided in Figure 9.

