# OpenReview forum: "Leveraging Per-Instance Privacy for Machine Unlearning"
_ICML.cc/2025/Conference — ICML 2025 poster_

### Official Review · Reviewer_1kMk · 2025-02-20

**Overall Recommendation:** 2

**Summary:**

This work introduces a theoretically grounded, data-dependent measure of difficulty between data privacy and unlearning. While the current analysis has limitations (scalability, group guarantees), the empirical results are compelling.

**Claims And Evidence:**

The identification of "harder" forget sets via privacy losses and the loss barrier analysis offer actionable insights for evaluating and improving unlearning algorithms.

**Essential References Not Discussed:**

Not any.

**Experimental Designs Or Analyses:**

How does varying the forget set size (e.g., 10% vs. 50% of data) affect the predictive power of privacy losses?

**Methods And Evaluation Criteria:**

Extensive experiments on SGLD, SGD, and L1-Sparse unlearning demonstrate the practical relevance of privacy losses. The inclusion of loss barrier analysis adds a geometric perspective to unlearning efficacy.

**Other Comments Or Suggestions:**

The paper emphasizes correlation with existing proxies but does not rigorously establish causality between privacy losses and unlearning difficulty. The correlation seems trivial and needs more analyzed, for example, from casual effect perspective. The Monte Carlo estimation of privacy losses via checkpoints may introduce bias.

**Other Strengths And Weaknesses:**

Weakness: 1) Experiments are limited to small-scale datasets (CIFAR-10, SVHN) and architectures (ResNet-18/ViT). The scalability to large models (e.g., LLMs) or massive datasets remains unverified. 2) The paper acknowledges that group unlearning guarantees are not tight but does not propose concrete steps to address this gap. This limits practical applicability for multi-point deletion. 3) For SGD experiments, privacy losses are computed under an assumed implicit noise (σ ≤ 0.1). While pragmatic, this lacks theoretical justification and may not hold in non-stochastic settings.

**Questions For Authors:**

How do loss barriers correlate with adversarial robustness or membership inference attack success? Do smaller barriers imply better security post-unlearning?  Were other data difficulty metrics (e.g., influence functions) considered? How do they compare to privacy losses in terms of cost and predictive power? How does the computational cost of estimating per-instance privacy losses scale with model size (e.g., transformers) and dataset size? Could gradient checkpointing or sampling mitigate this? Can you provide empirical/theoretical evidence that the implicit noise assumption in SGD holds? How sensitive are the results to the choice of σ?

**Relation To Broader Scientific Literature:**

The correlation between privacy losses and established proxies (e.g., C-Proxy, EL2N) bridges theory and heuristic approaches, grounding the work in prior literature.

**Theoretical Claims:**

The paper provides a novel theoretical framework for per-instance unlearning difficulty using Renyi divergence and differential privacy. By replacing worst-case DP bounds with per-instance privacy losses, it offers a more nuanced understanding of unlearning dynamics.

---

> ### Author Rebuttal · Authors · 2025-04-01
>
> Thank you for your feedback. Your review raises several important points: concerns about scalability to larger models and datasets, the practical implications of group-level unlearning, assumptions around noise in SGD, and the strength of causal interpretations in our analysis. You also asked about sensitivity to experimental parameters such as forget set size.
>
>
> We address these concerns below with additional experiments and clarifications. We show that privacy losses remain predictive across varying forget set sizes and are robust to assumptions on implicit SGD noise. Our method remains practical even when theoretical guarantees do not extend cleanly to group unlearning, as we demonstrate empirical effectiveness. Furthermore, we explain clearly why our causal interpretations are theoretically grounded, rather than simply correlational. Finally, we discuss computational cost in the context of scalability.
>
>
> **Experimental Designs Or Analyses.Unclear effect of forget set size**
>
>
>  We ran additional experiments varying forget set size and found that privacy losses consistently separate easy and hard sets, even with smaller subsets. As set size increases, differences become smaller: this is intuitive as the variance in average privacy loss gets smaller. Figures for forget set sizes 100, 5,000, and 10,000 are available at:
>  https://files.catbox.moe/l8ky5z.png,
>  https://files.catbox.moe/odaa0g.png,
>  https://files.catbox.moe/dqxk61.png.
>  We will include these results and discussion in the revised draft.
>
>
> **Other Strengths And Weaknesses 1. Lack of scalability to larger models and datasets**
>
>
> While more large-scale experiments would strengthen our claims, we already include several evaluations across multiple models and datasets (and SGLD noise). We’d appreciate clarification on what specifically the reviewer would want to understand by using larger models.
>
>
> **Other Strengths And Weaknesses 2.Group-level guarantees lack theoretical tightness**
>
>
> Our experiments show that averaging per-instance privacy losses still reliably ranks groups by unlearning difficulty (see Figure.1). Though empirical, this provides a promising direction for future theoretical development on group privacy, especially as the current group analysis is too loose.
>
>
> **Other Strengths And Weaknesses 3. Assumed noise in SGD lacks justification**
>
>
>  We ran experiments to test sensitivity to noise values used in estimating privacy losses. For SVHN with ResNet-18: (1) Spearman correlation between rankings at σ = 0.01 and σ = 0.001 was 0.70 (p = 0.0). (2) Between σ = 0.001 and σ = 0.0005 was 0.99 (p = 0.0). (3) Between σ = 0.0005 and σ = 0.0001 was 0.99 (p = 0.0). These results suggest that rankings are largely noise-invariant, as long as some noise is present. Past work has shown evidence of observable noise during training due to software and hardware non-determinism [1]. We will include this discussion in our revised version.
>
>
> [1] Jia, Hengrui, et al. "Proof-of-learning: Definitions and practice." 2021 IEEE Symposium on Security and Privacy (SP). IEEE, 2021.
>
>
> **Other Comments Or Suggestions. Correlation vs causality**
>
>
>  Our theory establishes a link between privacy loss and the number of unlearning steps, stating that small privacy loss means only a few steps are required to unlearn. We would appreciate clarification on what is meant by a “causal effect perspective,” as our goal is to measure difficulty, not to infer intervention outcomes. We welcome suggestions if there's a specific causal framework the reviewer believes is applicable here.
>
>
> **Questions For Authors. Scalability of computation and checkpointing**
>
>
>  Our method computes privacy loss using gradients from only 35 (out of total 150 training epochs giving 47,000 steps). In contrast, C-Proxy averages predictions over all 150 epochs of  checkpoints. Average Gradient Norm also averages gradients from all training epochs. EL2N is cheaper, but less predictive (see https://files.catbox.moe/uc54l5.png). Using only 35 checkpoints makes our method more efficient and scalable than most alternatives. We agree that techniques like gradient checkpointing or sampling could further reduce cost and plan to explore these in future work.

---

### Official Review · Reviewer_DFL7 · 2025-03-14

**Overall Recommendation:** 3

**Summary:**

This paper presents a per-data sample approach to quantifying the difficulty of unlearning the same via fine-tuning. They do this by adapting the definition and analysis of Thudi et al. (https://arxiv.org/abs/2307.00310) on per-instance DP to the unlearning setting to produce a quantity called a “privacy loss”, which measures the difficulty of unlearning a particular sample. This privacy loss can be straightforwardly used to get a tighter estimate on the number of finetuning steps needed to unlearn a sample. They validate the usefulness of privacy losses through experiments, and introduce “loss barriers” as another empirical measure for the difficulty of unlearning.

**Claims And Evidence:**

The paper proposes a per-sample privacy loss which they claim indicates the difficulty of unlearning the sample. They show theoretically that this privacy loss controls the finetuning steps needed to unlearn. They also show empirically that the privacy loss correlates with the number of steps needed to unlearn a set of items. These claims are supported by clear and convincing evidence.

**Essential References Not Discussed:**

Not that I know of

**Experimental Designs Or Analyses:**

The experimental designs made sense to me. They performed five experiments:
1) Testing that privacy loss corresponds to number of steps needed to unlearn
2) Testing that privacy loss corresponds to the difficulty of privacy violating attacks
3) Testing that loss barrier also corresponds to privacy loss
4) Testing that privacy loss corresponds to other known measures of the difficulty of unlearning
5) Testing that privacy loss is a better measure of difficulty of unlearning vs C-Proxy by showing it can select harder forget sets

All of the setups made sense and showed what they were supposed to show. I think experiment 5 could be expanded upon, as it only compares against C-Proxy.

**Methods And Evaluation Criteria:**

The methods and evaluation criteria make sense to me.

**Other Comments Or Suggestions:**

- Nit: figure 1 should explicitly write out what UA/MIA/GUS mean

**Other Strengths And Weaknesses:**

Strengths
- The paper is a very timely publication that combines the analysis of Chien et al. https://openreview.net/forum?id=3LKuC8rbyV which analyzes the number of steps needed to unlearn a set of data points using the DP framework, and the analysis of Thudi et al. (https://arxiv.org/abs/2307.00310) which introduces per-instance privacy. By combining these two, they can produce a tighter analysis of the difficulty of unlearning data samples.
- The claims are well-supported with theoretical analysis and experiments
- The paper produces a clear contribution to the literature: a new measure of the difficulty of unlearning data samples which appears to be the best, pending more experimental evidence.

Weaknesses
- The value of this paper in practice would be in showing that privacy losses are the most accurate way to measure the difficulty of unlearning. The only figure that tries to demonstrate this is Figure 3. However, it only compares against C-Proxy. There are other proxy metrics (as shown in Figure 2), so it is important to compare privacy losses against all of them to be sure of the value of privacy losses in practice. Related to this, it would be good to include a discussion of the relative cost of computing each of these proxies compared to their relative accuracy at predicting the difficulty of unlearning.
- From what I understand, the loss barrier computes the sharpness of the loss curve between two different models. The authors compute the sharpness of the loss between the oracle model (the perfectly unlearned model) and the model before/after unlearning. The experiments show that the sharper the curvature on a particular set of data is, the sharper the curvature is. This analysis is nice, but I'm wondering what the motivation for this is? Does this loss barrier work help us in any tangible way?

**Questions For Authors:**

1. Can you provide a more comprehensive experiment for Figure 3 comparing against the known ways to measure unlearning difficulty? It is critical to make this comparison comprehensive IMO.
2. What is the value of the loss barrier discussion?

**Relation To Broader Scientific Literature:**

The paper is related to the broader unlearning literature, and takes the differential privacy approach. The paper does a good job of discussing the related work.

**Theoretical Claims:**

The theoretical claims appear to make sense. However, it was hard to understand the privacy loss definition so I could not verify it rigorously.

---

> ### Author Rebuttal · Authors · 2025-04-01
>
> Thank you for your feedback. Your review raises two key concerns: the completeness of experimental comparisons between our proposed privacy loss and other proxy metrics, and the practical value of the newly introduced “loss barrier” measure. You also suggest clarifying definitions and improving the accessibility of the theoretical explanations.
>
>
> In response, we have expanded our experiments to include comparisons against EL2N and Average Gradient Norm, which further confirm that privacy loss is more effective in identifying difficult-to-unlearn examples. We also clarify the purpose and practical relevance of the loss barrier metric, showing that it provides an additional, independent way to evaluate unlearning efficacy. Each of your points is addressed below, underscoring that our proposed metrics provide robust, practical, and comprehensive methods for evaluating unlearning difficulty.
>
> **Theoretical Claims. Unclear definition of privacy loss**
>
>
> Could the reviewer clarify which parts of the definition were unclear? We recognize that the current form includes several moving parts, and we will revise the explanation to improve accessibility in the final draft.
>
>
> **Weaknesses 1. Need for broader comparison in Figure.3**
>
>
>  We agree. In addition to C-Proxy, we have now conducted comparisons with EL2N and Average Gradient Norm. Privacy loss continues to identify harder-to-unlearn examples than all alternatives. The new figure is available at https://files.catbox.moe/uc54l5.png and will be included in the revised draft.
>
>
> **Weaknesses 1. No discussion on relative computational cost of different proxies**
>
>
>  Thank you for raising this point. We computed privacy losses using 35 evenly spaced training checkpoints (from the 47,000 steps of training). All other methods—except EL2N—require access to all training checkpoints, making them more expensive. C-Proxy averages predictions across all checkpoints, while Average Gradient Norm uses gradients computed at each checkpoint. EL2N is cheaper but less effective, as shown in the new comparison figure linked earlier.
>
>
> **Weaknesses 2. Unclear motivation behind the loss barrier metric**
>
>
> The loss barrier provides an additional way to assess unlearning, providing a novel membership inference (MI) attack. Specifically, discrepancies from the expected loss barrier between two oracle models can indicate whether unlearning was successful. This provides a novel way to assess whether the model has achieved the correct loss geometry after unlearning—something not captured by existing metrics, which focus only on changes in output or loss at individual points. Its contribution to the paper is to offer another metric to evaluate our main claim: that privacy losses accurately predict unlearning difficulty.
>
>
> **Other Comments Or Suggestions. Clarify UA/MIA/GUS in Figure.1**
>
>
>  Thank you—we will revise the caption in Figure.1 to clearly define these terms.

---

### Official Review · Reviewer_R2F4 · 2025-03-14

**Overall Recommendation:** 3

**Summary:**

The paper proposes a per-instance approach to quantifying the difficulty of unlearning via fine-tuning by replacing the worst-case Rényi-DP bound with per-instance privacy losses. The authors introduce loss barriers as a way for evaluation, which are significantly reduced after unlearning. Alternative cheap proxy measures of data difficulty are explored for better efficiency. Empirical results demonstrate that the privacy losses offer a precise and actionable measure of unlearning difficulty and could identify harder data.

**Claims And Evidence:**

Yes.

**Essential References Not Discussed:**

None.

**Experimental Designs Or Analyses:**

The reviewer has checked all of the experimental designs.

**Methods And Evaluation Criteria:**

Yes.

**Other Comments Or Suggestions:**

Using figure illustrations aiding analysis in Section 4 and Section 5.2 would be better, as done in Chien et al,. 2024.

**Other Strengths And Weaknesses:**

Strengths:

1. The paper innovatively introduces per-instance privacy losses for unlearning difficulty measurement and provides theoretical unlearning guarantees.
2. The paper is well-structured. The narrative is easy to follow.

Weaknesses:

1. Computing per-instance privacy losses requires storing and processing gradients throughout training, which can be computationally expensive, especially for large-scale models and datasets. This limits the practicality of the approach. The reviewer supposes that the proposed method can serve as a strategy for model pruning before unlearning, which can then be used to design experiments on more complex datasets and models.

**Questions For Authors:**

In the caption of Figure 1, what do you mean by "Baseline corresponds to the loss barrier between two oracles"? What are the "two oracles" here?

**Relation To Broader Scientific Literature:**

This paper presents work whose goal is to advance the field of machine unlearning, which is specifically oriented to improve the trustworthiness of machine learning.

**Theoretical Claims:**

The reviewer did not check the correctness of the proofs.

---

> ### Author Rebuttal · Authors · 2025-04-01
>
> Thank you for your feedback. Your review focuses on question around the computational efficiency of our method for computing per-instance privacy losses, especially in the context of large-scale models and datasets. You also provide suggestions for improving the clarity of figure and definitions, which we address below.
>
>
> Perhaps the most important statement for us to make about efficiency is that _our approach is significantly more efficient than all alternatives we study__ with the exception of EL2N which is inferior. We require gradients from only 35 checkpoints from the 150 training epochs. At present, alternative methods currently rely on all checkpoints. We also appreciate your suggestions for improving our illustrations and definitions, and we have incorporated them into our planned revisions. Our detailed responses aim to reinforce a key point: our method efficiently and reliably quantifies unlearning difficulty, maintaining practical feasibility even for larger models and datasets.
>
>
> **Weaknesses. Computational efficiency and practical applicability**
>
>
> To clarify, our method requires storing 35 checkpoints and computing gradients on them for the datapoints being evaluated. Given that our full training run consists of 150 epochs, the  estimation of per-instance guarantees for the whole dataset costs about 20% of training time. Other methods, such as C-Proxy and Average Gradient Norm, are more expensive: C-Proxy averages predicted probabilities across all 150 training checkpoints, and Average Gradient Norm computes gradients at every checkpoint. In contrast, we use only 35 evenly spaced checkpoints. While EL2N is cheaper, it performs worse in identifying hard-to-unlearn examples (see https://files.catbox.moe/uc54l5.png).
> Regarding the idea of using privacy loss for model pruning, we weren’t entirely sure what the reviewer meant—could you clarify this suggestion further?
>
>
> **Other Comments Or Suggestions. Using figure illustrations**
>
>
>  Thank you for the helpful suggestion. We will include a similar diagram in the revised draft, following Chien et al. (2024), but highlighting that initial divergence varies across datapoints and that many datapoints begin with near-zero divergence---indicating they require almost no steps to unlearn.
>
>
> **Questions For Authors. Clarification on Figure.1 caption**
>
>
>  We appreciate the chance to clarify this. Due to stochasticity in training (e.g., minibatch sampling, GPU nondeterminism), two runs on the same dataset can result in slightly different models. To control for this noise in loss barrier evaluations, we compute the loss barrier between two models trained independently on the same retain dataset. These serve as “oracles” representing ideal retraining, and the baseline corresponds to the loss barrier between them. This gives us a reference point for how small the barrier can be when unlearning is perfect.

---

### Official Review · Reviewer_PpYB · 2025-03-15

**Overall Recommendation:** 3

**Summary:**

This paper provides a theoretical foundation to understand the relationship between per-instance privacy loss and unlearning hardness. It applies a recent per-instance privacy loss to fine-tuning-based unlearning and builds a relationship bewteen unlearning steps with the bound of Renyi divergence between a model trained with and without an individual data point. They then empirically show the relationship of the per-instance privacy loss to approximate unlearning steps.

**Claims And Evidence:**

The paper’s contributions are primarily theoretical. While proposing a novel metric for per-instance unlearning difficulty, it lacks evidence of practical impact (e.g., would unlearning-via-fine-tuning methods with the proposed privacy loss perform better than relying on gradient norms?). Stronger empirical validation (e.g., comparative experiments) is needed to demonstrate real-world utility.

**Essential References Not Discussed:**

N/A

**Experimental Designs Or Analyses:**

- All experiments focus on group unlearning; however, the theoretical analysis provides guarantees only for per-instance scenarios. Including experiments demonstrating ​per-instance privacy loss alignment to unlearning steps would strengthen the practical relevance of the theoretical claims.
- ​Figure 2: The authors claim that existing proxy privacy losses overestimate unlearning difficulty, but this may stem from scaling differences (e.g., the per-instance hardness  is logarithmic with respect to the proposed loss). A fairer comparison is required. (Figure 3 partially addresses this but only contrasts with C-Proxy, which appears to underperform in Figure 2).

**Methods And Evaluation Criteria:**

Yes.

**Other Comments Or Suggestions:**

- if you could provide the distribution overlap between the constructed forget set and retain set or show any identified hard-to-forget examples, it would be helpful to understand the metric.

**Other Strengths And Weaknesses:**

N/A

**Questions For Authors:**

- I’m curious if we could use the proposed privacy loss to enhance unlearning performance or reduce unlearning time, like what Zhao et al. (2024) did. Could you show me how?

**Relation To Broader Scientific Literature:**

This work establishes a theoretical foundation to support prior observations that specific forget sets exhibit inherent challenges during unlearning.

**Theoretical Claims:**

This paper provides a better convergence analysis of $U_{k}(\alpha, \epsilon^*)$-Renyi unlearning. By directly applying the per-instance privacy loss from Thudi et al. ( 2024) into the unlearning setting, this paper builds the bound of Renyi divergence between a model trained with and without an individual data point. Then it builds a connection between the unlearning steps and the Renyi divergence bounds, which is bounded by the per-instance privacy loss. Then it is able to use per-instance privacy loss to predict per-instance unlearning hardness.

---

> ### Author Rebuttal · Authors · 2025-04-01
>
> Thank you for your thoughtful feedback. Your review primarily raises concerns about the practical utility of our proposed privacy loss metric—specifically, whether its per-instance theoretical guarantees translate to improved empirical unlearning performance. You also highlight the focus on group-level experiments and question whether our comparisons with existing metrics fairly demonstrate superiority. We appreciate these comments and have expanded our response and experiments accordingly.
>
>
> As outlined below, we clarify that our empirical results already show practical value, particularly by demonstrating that examples with low privacy loss consistently require near 0 unlearning steps. We also include new comparisons against EL2N and Average Gradient Norm, which confirm that privacy loss better identifies harder-to-unlearn datapoints. Taken together, our theoretical and empirical results reinforce the claim that our proposed privacy loss metric is valuable for explaining unlearning performance.
>
>
> **Claims And Evidence. Lack of evidence for practical impact**
>
>
>  Figure.1 demonstrates that datapoints identified as easy by privacy loss require near-zero steps to unlearn with fine-tuning. We believe this already shows practical impact: for examples in the easiest quantiles, unlearning can be achieved with just a few fine-tuning steps. This allows privacy loss to serve as a signal for when minimal unlearning is sufficient—something existing metrics do not support as reliably.
>
>
> **Experimental Designs Or Analyses 1. Including experiments demonstrating ​per-instance privacy loss alignment to unlearning steps**
>
>
> This alignment is already shown in Figure.1, where we demonstrate that privacy loss ranks groups of datapoints by the number of steps needed to unlearn them. Our group-level results are derived by averaging per-instance privacy losses, which provides a straightforward way to extend pre-computed individual guarantees. We also tested Thudi et al.’s group privacy analysis and found it too loose to distinguish between easy and hard groups (Appendix B), further highlighting the value of our empirical approach. Future work on group privacy could study the correctness/discrepancy with averaging per-instance data points to improve the current group privacy theory.
>
>
> **Experimental Designs Or Analyses 2. Claim about overestimation in Figure.2**
>
>
> We agree that the original claim in Figure.2 was misleading. We now clarify that Figure.2 shows a correlation, but not necessarily overestimation due to scaling mismatches. The overestimation claim is instead supported by Figure.3, which demonstrates that datapoints identified as “hard” by other metrics are in fact easier to unlearn than those identified as hard by privacy loss. To further support this point, we’ve added new experiments comparing privacy loss against EL2N and Average Gradient Norm. Privacy loss continues to identify harder examples more effectively. These results are available at https://files.catbox.moe/uc54l5.png and will be added to our revised draft.
>
> **Other Comments Or Suggestions. Request for overlap analysis or examples**
>
>
> We have included a qualitative example showing four randomly selected images from both an easy-to-forget set (https://files.catbox.moe/2qdv6h.jpg) and a hard-to-forget set (https://files.catbox.moe/tzm6ac.jpg), each of size 1000 and constructed from the CIFAR-10 dataset. Regarding the distribution overlap, we would be happy to provide it—could the reviewer kindly clarify which specific distribution they are referring to?
>
>
> **Questions For Authors. Could privacy loss help reduce unlearning time?**
>
>
>  Yes—this is already possible using our results. For instance, in Figure.1 we show that the easiest 50% of datapoints (as measured by privacy loss) can be unlearned in fewer than five fine-tuning steps. This suggests a simple strategy: apply lightweight fine-tuning for low-privacy-loss points, and use stronger methods (e.g., retraining or more targeted interventions) for datapoints flagged as difficult. This could easily be integrated into frameworks like RUM [1], where privacy loss can guide refinement.
>
>
> [1] Zhao, Kairan, et al. "What makes unlearning hard and what to do about it." Advances in Neural Information Processing Systems 37 (2024): 12293-12333.

---

### Official Review · Reviewer_T3UX · 2025-03-18

**Overall Recommendation:** 3

**Summary:**

This paper considers the machine unlearning problem, which involves removing the influence of a subset of training data from a trained model. The authors explored a setup in which both learning and unlearning are done via noisy gradient descent and proposed to use the "per-instance privacy loss" to estimate the unlearning difficulty.  theoretically demonstrate that the number of unlearning rounds depends at most logarithmically on the "per-instance privacy loss" (Corollary 4.4).  Additionally, the authors presented extensive experimental results indicating that "per-instance privacy loss" provides a more accurate estimation of unlearning difficulty (i.e., time to unlearn) compared to existing proxy metrics, and it effectively identifies examples that are challenging to unlearn.

**Claims And Evidence:**

1. The authors stated in line 401 that "Figure 2 reveals that all these proxies overestimate the difficulty." However, Figure 2 only presents a comparison of proxies against privacy loss and does not directly relate to unlearning difficulty. To substantiate the claim that a metric overestimates difficulty, the authors should provide evidence showing that data points with relatively high metric values can be unlearned in the same (or even fewer) number of rounds as those with lower values.

2. In Section 6, the authors asserted that "privacy losses accurately predict the number of unlearning steps." While the empirical results indicate that the time to unlearn increases with privacy loss, this alone does not justify the term "accurately," as there is no established predictive rule correlating privacy loss to the exact number of unlearning rounds.

3. In Section 6.4, the authors compared privacy loss only with the C-Proxy. It is unclear why they did not include comparisons with other proxy metrics mentioned in Section 6.3. Are those other metrics demonstrated to be less effective than C-Proxy in previous work?

**Essential References Not Discussed:**

N/A

**Experimental Designs Or Analyses:**

The authors provided comprehensive experiments to support their proposed "per-instance privacy loss".

1. I have raised some questions regarding the claims made based on the experimental results in the "Claims and Evidence" section.

2. I would like to inquire about the number of checkpoints ($N$) used in estimating the per-instance privacy loss.

3. While the experiments demonstrate that privacy losses can identify data that are hard to unlearn, do they also indicate which data are easy to unlearn?

**Methods And Evaluation Criteria:**

The learning and unlearning algorithms, as well as the datasets used, are appropriate for addressing the unlearning problem discussed in the article.

**Other Comments Or Suggestions:**

The definition of "5% margin" in the caption of Figure 1 is not sufficiently clear. I recommend using the definition provided in Figure 6, which is more precise, as it specifies that "5% margin" means the difference between the unlearned model’s UA and the oracle’s UA for the given forget set.

**Other Strengths And Weaknesses:**

Strengths:

1. The idea of employing the Loss Landscape to demonstrate how geometric properties reflect unlearning difficulty is intriguing.

2. The authors conducted comprehensive experiments that effectively support their proposed method.

Weaknesses:

1. The proposed privacy loss may not accurately estimate unlearning difficulty for groups, as it focuses on per-instance loss and does not account for correlations within the group.

2. Estimating the per-instance privacy loss could be resource-intensive, as it requires gradients from multiple checkpoints.

**Questions For Authors:**

I would appreciate it if the authors could address my concerns raised in the "Claims and Evidence" section.

**Relation To Broader Scientific Literature:**

Unlearning has significant implications for user privacy, as it enables users to delete their data from the trained model, and it can contribute to the development of more robust machine learning models by removing malicious examples. Additionally, the proposed technique may serve as a method for detecting outliers within datasets.

**Theoretical Claims:**

I did not conduct a thorough review of the proofs, but they appear to be correct.

---

> ### Author Rebuttal · Authors · 2025-03-31
>
> Thank you for your thoughtful feedback. Your review raises two main concerns: (1) the completeness of comparisons with alternative proxy metrics, and (2) the clarity of our empirical claims regarding how accurately privacy losses predict unlearning difficulty. To address the first concern, we have expanded our experiments to include comparisons against additional proxies—EL2N and Average Gradient Norm. These comparisons (available at https://files.catbox.moe/uc54l5.png) support our original finding: privacy loss consistently identifies harder-to-unlearn examples than existing metrics.
>
>
> Regarding the second concern about the clarity of our empirical claims, we believe the ambiguities arise from our dense discussion in our original draft. Below, we describe how we have revised the text to clarify that our claims are already supported by existing figures---e.g., Figure.1 shows privacy losses can rank points by number of unlearning steps, while Figure.3 compares the "hardness" of C-proxy and our approach, as measured by training steps to unlearn. We have also clarified definitions where needed. Below we respond to individual points raised in the review.
>
>
> **Claims And Evidence 1. Figure.2 does not directly relate to unlearning difficulty**
>
>
> We have revised the text for clarity: Figure.2 demonstrates that past metrics correlate well with privacy losses. When discussing "overestimating difficulty" we now clearly refer to Figure.3 in our paper, which shows the hardest points identified by C-proxy take fewer steps to unlearn than the hardest points identified by privacy losses. Furthermore, we are updating Figure.3 to also include EL2N and Average Gradient Norm; we found privacy losses still identify the harder datapoints. See updated results at https://files.catbox.moe/uc54l5.png.
>
>
> **Claims And Evidence 2. No predictive rule correlating privacy losses and the number of unlearning steps**
>
>
> We appreciate this point. Our intended claim is not that privacy losses predict the exact number of steps, but rather that they accurately rank datapoints by how long they take to unlearn. This is shown in Figure.1, where unlearning time increases with privacy loss. We will revise the language to say: “privacy losses accurately rank datapoints according to the number of steps needed to unlearn.”
>
>
> **Claims And Evidence 3. Compared privacy loss only with C-Proxy In Figure.3**
>
>
> As mentioned above, we have now included EL2N and Average Gradient Norm in our experiments. Privacy loss remains more effective at identifying hard-to-unlearn examples.
>
>
> **Experimental Designs Or Analyses 2.Number of checkpoints (N) used in estimating privacy loss**
>
>
> We used 35 checkpoints, evenly spaced across the 150 training epochs (47,000 steps). We will add this clarification.
>
>
> **Experimental Designs Or Analyses 3.Do privacy losses also identify easy-to-unlearn data?**
>
>
> Yes. As shown in Figure.1, datapoints with the lowest privacy loss scores require virtually no unlearning steps. More broadly, privacy loss correlates with unlearning time across the full range of scores—higher privacy loss values generally correspond to more steps required to forget a sample. This pattern is shown in Figure.1 (middle panel) and is consistent with the trend observed in Figure.5.
>
>
> **Weaknesses 1. privacy loss does not accurately estimate unlearning difficulty for groups**
>
>
> We appreciate this concern. In Appendix B, we evaluated Thudi et al.’s group-based method and found it failed to distinguish between easy and hard-to-unlearn subsets. In contrast, simply averaging our per-instance privacy losses did capture group-level difficulty (see Figure.1). We believe this result is both practical and empirically sound—and it highlights a useful direction for theory to catch up with practice in deep learning.
>
>
> **Weaknesses 2. Estimating privacy loss may be resource-intensive**
>
>
> We estimate privacy losses using gradients from just 35 of the 47,000 training steps—only a small fraction of the training cost. By contrast, other metrics like C-Proxy and Average Gradient Norm rely on access to every training checkpoint (150 epochs). EL2N avoids this but performs worse. So our method strikes a good balance between efficiency and predictive power.
>
>
> **Other Comments Or Suggestions. "5% margin" in caption of Figure.1 is unclear**
> Thanks for pointing this out. We will clarify that the "5% margin" refers to the value of the unlearning metric (e.g., UA or MIA) measured on the oracle model: i.e., we're within 5% of the oracle UA (or MIA), depending on the metric being evaluated.

---

> > ### Comment · Reviewer_T3UX · 2025-04-03
> >
> > Thank you for your clarification and additional result. The privacy loss does outperform other proxies and achieve a efficiency vs predictive precision tradeoff. I have increased my score to 3.
> >
> > Just a minor issue about the word choice: I still think the word "overestimate" is a bit misleading, as it usually refers to obtaining something higher than the ground truth. In the context of identifying hard-to-unlearn examples, though the past metrics do classify some easy-to-unlearn examples as hard-to-learn, they also at the same time classify some hard-to-unlearn examples as easy-to-learn. This behavior is more analogous to misclassification than overestimation. That's why I felt confused when reviewing the article. Maybe something like "inaccurate" is a better choice than "overestimate".

---

> > > ### Author Response · Authors · 2025-04-03
> > >
> > > Thank you for your response!
> > >
> > > > I still think the word "overestimate" is a bit misleading, as it usually refers to obtaining something higher than the ground truth. In the context of identifying hard-to-unlearn examples, though the past metrics do classify some easy-to-unlearn examples as hard-to-learn, they also at the same time classify some hard-to-unlearn examples as easy-to-learn. This behavior is more analogous to misclassification than overestimation. That's why I felt confused when reviewing the article. Maybe something like "inaccurate" is a better choice than "overestimate".
> > >
> > > Sorry for missing this, yes we completely agree. We will change the use of “overestimating” to “inaccurate” throughout the paper. Thanks for the suggestion!
> > >
> > > If you would permit us to be bold, if you think the paper should be accepted, we would ask you to vote "Accept", because at the moment, all of our votes are borderline, which is going to leave the decision up to the wind.

---

### Decision · Program_Chairs · 2025-05-01

**Decision:**

Accept (poster)

**Comment:**

This paper presents a theoretically grounded and empirically validated framework for quantifying per-instance unlearning difficulty via privacy loss, with connections to Rényi divergence and DP guarantees. The paper is well-organized, with clear claims and extensive experiments demonstrating that privacy loss can effectively rank examples by unlearning difficulty.

Reviewer consensus generally leaned positive. The reviewers appreciated the novelty of per-instance privacy loss as a theoretically justified metric for unlearning hardness, and the extensive empirical evaluation across unlearning setups such as SGD, SGLD, L1-sparse fine-tuning etc. Reviewers also valued the addition of loss barrier analysis as a complementary geometric perspective.

There were also some reasonable concerns raised. The main critique is the limited practical evidence that privacy loss enables faster or more efficient unlearning in real-world systems. Reviewers also asked for a more comprehensive comparison against existing proxy metrics (EL2N, Average Gradient Norm), which the authors addressed convincingly in the rebuttal with new experiments. The cost of estimating privacy loss remains non-trivial (35 checkpoints), although the authors argue it is cheaper than alternatives like C-Proxy.

Overall, the paper makes a clear, well-supported theoretical and empirical contribution to the growing literature on machine unlearning. While not all practical challenges are resolved, the work is timely, rigorous, and likely to influence future research.